# Synergistic Interactions of 5-Fluorouracil with Inhibitors of Protein Kinase CK2 Correlate with p38 MAPK Activation and FAK Inhibition in the Triple-Negative Breast Cancer Cell Line

**DOI:** 10.3390/ijms21176234

**Published:** 2020-08-28

**Authors:** Patrycja Wińska, Olena Karatsai, Monika Staniszewska, Mirosława Koronkiewicz, Konrad Chojnacki, Maria Jolanta Rędowicz

**Affiliations:** 1Chair of Drug and Cosmetics Biotechnology, Faculty of Chemistry, Warsaw University of Technology, 00-664 Warsaw, Poland; mstaniszewska@ch.pw.edu.pl (M.S.); kchojnacki@ch.pw.edu.pl (K.C.); 2Laboratory of Molecular Basis of Cell Motility, Nencki Institute of Experimental Biology, Polish Academy of Sciences, 02-093 Warsaw, Poland; o.karatsai@nencki.edu.pl (O.K.); m.redowicz@nencki.gov.pl (M.J.R.); 3Department of Drug Biotechnology and Bioinformatics, National Medicines Institute, 00-725 Warsaw, Poland; m.koronkiewicz@nil.gov.pl

**Keywords:** 5-fluorouracil, apoptosis, breast cancer, FAK, p38 MAPK, protein kinase CK2, synergism

## Abstract

Background: The combination effect of 5-fluorouracil (5-FU) with either CX-4945 or a new inhibitor of protein kinase CK2, namely 14B (4,5,6,7-tetrabromo-1-(3-bromopropyl)-2-methyl-1*H*-benzimidazole), on the viability of MCF-7 and triple-negative MDA-MB-231 breast cancer cell lines was studied. Methods: Combination index (CI) values were determined using an MTT-based assay and the Chou-Talalay model. The effect of the tested drug combinations on pro-apoptotic properties and cell cycle progression was examined using flow cytometry. The activation of FAK, p38 MAPK, and ERK1/2 kinases and the expression of selected pro-apoptotic markers in MDA-MB-231 cell line after the combined treatment were evaluated by the western blot method. Confocal microscopy was used to examine actin network in MDA-MB-231. Results: Our results showed that a synergistic effect (CI < 1) occurred in MDA-MB-231 after treatment with both combinations of 5-FU with 14B or CX-4945, whereas the combination of 5-FU and 14B evoked an antagonistic effect in MCF-7. We conclude that the synergistic interactions (CI < 1) observed for both the combinations of 5-FU and 14B or CX-4945 in MDA-MB-231 correlated with an activation of p38 MAPK, inhibition of FAK, increased expression of apoptogenic markers, prolongation of S-phase of cell cycle, and destabilization of actin network. Conclusions: The obtained results support the recent observation that CK2 inhibitors can improve 5-FU-based anticancer therapy and FAK kinase can be an attractive molecular target in breast cancer therapy.

## 1. Introduction

Breast cancer is the most common malignancy among women worldwide and is the second (after lung cancer) most frequent cause of death in women [1]. Breast cancer is categorized into three major subtypes based on the presence or absence of molecular markers for estrogen or progesterone receptors and human epidermal growth factor 2 (ERBB2, formerly HER2): (i) hormone-receptor-positive/ERBB2-negative (70% of patients), (ii) ERBB2-positive (15–20%), and (iii) triple-negative (tumors lacking all three standard molecular markers; 15%) [2]. Depending on their receptor signature, these breast cancer subtypes have distinct neoadjuvant/adjuvant treatments, such as hormonal agents and cytotoxic chemotherapy either simultaneously or consecutively [3]. Patients with triple-negative breast cancer (TNBC) are difficult to treat by hormonal or anti-HER2 therapy, and only classical cytotoxic agents, such as 5-fluorouracil (5-FU), offer a viable option for those who develop distant metastasis [4,5]. 5-FU is converted inside cells into 5-fluoro-dUMP (F-dUMP, fluoro-deoxyuridine monophosphate), which forms a stable complex with thymidylate synthase (TS) and thus inhibits deoxythymidine monophosphate (dTMP) production that is essential for DNA replication and repair [6]. Another important mechanism of 5-FU-induced cytotoxicity is its misincorporation into RNA and DNA in the place of uracil or thymine that consequently activates DNA repair and finally double-strand break (DSB) induction and DNA damage [7]. Due to its cytotoxicity [8] and resistance occurrence, 5-FU is one of the most commonly used TS-directed anticancer drugs applied in combination with other chemotherapy compounds for the treatment of various cancers, including breast cancer [9,10].

Our recent studies demonstrated a synergistic interaction of 5-FU and a clinical stage inhibitor of protein kinase CK2, CX-4945, in hormone-dependent MCF-7 breast cancer [11]. The synergistic interactions were related to the delay of 5-FU-induced S-phase arrest recovery, leading to 17% more cells in S-phase versus control after 72 h, compared to 48 h of treatment. CK2 belongs to a serine/threonine kinase family and phosphorylates numerous proteins associated with the regulation of many signal pathways involved in cell cycle regulation, cell proliferation, and apoptosis, such as cell division control protein 37 (CDC37), phosphatase and tensin homolog deleted on chromosome ten (PTEN), p53 tumor protein, transcription factor p65 (NF-κB subunit), IKAROS family zinc finger 1 (IKFZ1), caspases 2 and 9, and cytoskeleton proteins [12]. The overexpression of CK2 is frequently detected in a variety of cancers including prostate, breast, and lung cancers [13]. Moreover, several evidences support a role for CK2 in the processes directly responsible for drug resistance, such as drug efflux and DNA repair. CK2 intervenes in signaling pathways which are crucial to evade drug response (i.e., PI3K, phosphoinositide 3-kinase/AKT/PTEN, NF-κB, β-catenin, hedgehog signaling, and p53), and controls the activity of chaperone machineries fundamental in resistant cells [14].

Many studies demonstrated that inhibition of CK2 by small ATP-competitive inhibitors such as 4,5,6,7-tetrabromo-1*H*-benzimidazole (TBBi) and its derivatives, or by CX-4945 (silmitasertib) which caused a range of phenotypic changes in breast cancer cell lines, including decreased viability, cell cycle arrest, apoptosis, and loss of migratory capacity [15,16]. The inhibition of CK2 by CX-4945 caused cell cycle arrest in G2–M transition in breast cancer cells, and activated caspase 3 and caspase 7 in cancer cells with no detectable change of caspase 3/7 activity in normal cells. In vitro studies employing CX-4945 in combination with gemcitabine or cisplatin revealed enhanced antiproliferative effects in A2780 and SKOV-3 ovarian cancer cells. The antiproliferative activity was 23–38% higher than Bliss additivity when CX-4945 was added to cells after treatment with gemcitabine or cisplatin. These treatments were also accompanied by decreased phosphorylation of the CK2 substrates, XRCC1 and MDC1, and accumulation of single-stranded and double-stranded DNA breaks [17]. Interestingly, the most recent data demonstrated that a combination of CK2α knockdown with 5-FU treatment promoted apoptosis of 5-FU-resistant colorectal cancer (CRC) cells by inducing ER (endoplasmic reticulum) stress, pointing at CK2α inhibition as a promising adjuvant therapeutic approach in combination with 5-FU for CRC [18].

Our present study aimed to investigate the combination effect of 5-FU with either CX-4945 or a new inhibitor of protein kinase CK2, namely 14B (4,5,6,7-tetrabromo-1-(3-bromopropyl)-2-methyl-1*H*-benzimidazole), on the viability of triple-negative MDA-MB-231 breast cancer cells. We have previously demonstrated that the newly obtained TBBi derivative, 14B, inhibited CK2α catalytic subunit and CK2 holoenzyme with *K_i_* values of 0.82 µM and 0.18 µM, respectively, and affected MCF-7 viability in a low micromolar value (EC_50_ = 3.4 µM) [19]. To evaluate the combination effect of simultaneous treatment of breast cancer with 5-FU and CK2 inhibitors, we sought to determine combination index (CI) values in the two breast cancer cell lines, MCF-7 and MDA-MB-231. Our results showed that a synergistic effect (CI < 1) occurred in MDA-MB-231 after treatment with both the combinations of 5-FU with 14B or CX-4945, whereas a combination of 5-FU and 14B evoked an antagonistic effect in MCF-7. To explain the molecular mechanism of the observed synergistic effect, the influence of the tested combinations on pro-apoptotic properties, cell cycle progression, CK2 inhibition (phosphorylation extent of Ser529 p65), TS and CK2α protein level changes, and other protein kinases (i.e., FAK, focal adhesion kinase, p38 MAPK, and ERK1/2) were examined in MDA-MB-231 cells. 

## 2. Results

Two types of breast cancer cell lines, i.e., triple-negative MDA-MB-231 and hormone-dependent MCF-7, were treated with the combinations of 5-FU and one of the inhibitors of CK2 (CX-4945 or the recently obtained 14B). Among these compounds, CX-4945 is in stage I/II of clinical trials, 5-FU is a well-known prodrug targeting TS, whereas 14B was recently synthesized in our department [19] as a new compound which efficiently induced inhibition of CK2 in MCF-7 and demonstrated better anticancer properties against MCF-7 than its parent compound TBBi. An MTT-based assay and the combination index (CI) method [20] were used to determine the type of interaction (i.e., whether it could be synergistic, additive, or antagonistic) between one of the CK2 inhibitors (14B or CX-4945) and 5-FU (inhibition of TS by the 5-FU metabolite, F-dUMP). Additionally, the dose reduction index (DRI) was calculated on the basis of a drug interaction data analysis. This parameter is inversely associated with CI and represents the number of times each single drug dose may be reduced in a combination setting without compromising the final therapeutic effect [20].

### 2.1. Compounds’ Influence on the Viability of MDA-MB-231 and MCF-7 Cell Lines

To optimize the ratio of the compounds used in the combination treatment, the influence on the cell viability of each compound when used alone was determined by obtaining *D_m_* values, describing the drug potency. The results are summarized in Table 1. Among the tested compounds, the lowest *D_m_* values were obtained for both the studied lines for the new derivative of TBBi, 14B, with very similar values of 3.94 ± 1.08 µM and 4.28 ± 0.56 µM for MDA-MB-231 and MCF-7 lines, respectively (Table 1). Interestingly, the significant difference in 5-FU potency was detected for the two types of the studied breast cancer lines, with the *D_m_* values more than 4 times higher for MDA-MB-231 than for MCF-7. The ratio of the test compounds used in the combinations, specified by their *D_m_* values and also by the preliminary results (data not shown) provided the fraction of not viable cells (Fa) in the range of 0–1. Six to eight concentrations of each compound, in the range from 0.125 × *D_m_* to 6 × *D_m_* in a constant ratio at 2-fold dilution series according to recommendations given by Chou [20], were used in combination experiments. Combination index (CI) values were generated in CalcuSyn Software at ED_50_, ED_75_, and ED_90_ after fitting Fa values obtained by the MTT-based assay (Table 2).

As it is important to reduce toxicity in clinical treatment and 5-FU is an essential component of chemotherapy in the breast cancer and is commonly used in consolidation therapy, dose reduction index (DRI) values were shown for this drug when tested in combinations. The obtained CI values for MDA-MB-231 were in the range of 0.64–0.76, thus indicating synergism. The obtained data indicated that the dose of 5-FU in a combination with the 14B derivative or CX-4945 may be decreased by about 43-fold and 46-fold, respectively, at 95% of affected MDA-MB-231 cells. Representative examples of the graphs generated from the analysis of two compound combinations, demonstrating the synergistic effect of 5-FU in combination with 14B or CX-4945 in MDA-MB-231 cells, are shown in Figure 1. In contrary to MDA-MB-231 and the previously described synergistic effect in MCF-7 after treatment with 5-FU and CX-4945 [11], antagonism was observed for this line after treatment with the combination of 5-FU and 14B with CI values in the range of 1.16–2.94. Therefore, more detailed studies were conducted for combinations of 5-FU with either 14B or CX-4945 in MDA-MB-231.

The influence of simultaneous treatment of MDA-MB-231 with 5-FU and either 14B or CX-4945 combinations on cell cycle progression, apoptosis, and cellular level of TS and CK2α proteins as well as on the activity of CK2 were investigated. Additionally, an activation of the selected signaling pathways, including FAK, p38, and ERK1/2 kinases, were studied. Due to the strong synergistic effect observed in MDA-MB-231 after treatment with both the combinations, the following concentrations of inhibitors were used in subsequent assays: 0.25 × *D_m_*, 0.5 × *D_m_*, and 0.4 × *D_m_* for 5-FU, 14B, and CX-4945, respectively.

### 2.2. The Effect of Drug Combinations on TS, CK2α, and NF-κB-p65 in MDA-MB-231 Cells

In view of the observations that combinations of 5-FU with 14B or CX-4945 affect the viability of MDA-MB-231 in a synergistic manner, we examined the influence of these compounds used either separately or in combinations on TS and CK2α protein levels in cellular extracts. Additionally, the level of CK2-mediated phosphorylation of NF-κB-p65 was studied. Decreased phosphorylation of p65 was detected only after 48 h of treatment with 14B alone, 5-FU in combination with 14B, and CX-4945 alone with the relative expression values of 0.67, 0.5, and 0.88, respectively. Unexpectedly, the phosphorylation level of p65 on Ser529 was the highest in 5-FU-treated cells with up to 2 times the relative expression after 72 h treatment (Figure 2). Moreover, no inhibition was detected in cells treated with 14B or CX-4945 after 72 h of treatment. A partial correlation between p65 phosphorylation and CK2α level was observed (Figure 2B), as the level of CK2α was elevated in 5-FU-treated cells after 72 h, similarly to p-Ser529-p65. The use of higher concentrations of CK2 inhibitors resulted in insufficient cell viability in combinations to be tested. An effective inhibition of CK2 in MDA-MB-231 treated with 14B or CX-4945 can be difficult because of CK2α and CK2α’ (catalytic subunits) overexpression and their elevated activity in this cell line [15]. However, inhibition of CK2 activity in this line after treatment with 5 µM CX-4945 has been previously observed in special culture conditions, i.e., in serum-free medium [15].

The complex of TS with 5-fluoro-dUMP (the upper line in TS, Figure 2A), indicating TS inhibition in MDA-MB-231 cells, was detected after 5-FU treatment, as well as after treatment with a combination of 5-FU with either 14B or CX-4945. Since the obtained results were inefficient to conclude about the molecular mechanism of the observed synergistic interactions, additional signaling pathways were studied.

### 2.3. The Effect of Drug Combinations on p38 MAPK, FAK, and ERK1/2 Kinases in MDA-MB-231 Cells

To explain the molecular mechanism of the observed synergistic interactions of 5-FU with either 14B or CX-4945 in MDA-MB-231 cells, the activation of FAK, p38, and ERK1/2 kinases were evaluated after treatment with 5-FU, 14B, and CX-4945 used separately and in combinations. The most significant effect among both the tested combinations occurred in the p38 pathway after 5-FU and 14B treatment, especially after 72 h of treatment. The strongest activation of p38 kinase was observed in cells treated with 5-FU plus 14B after 48 h and 72 h of incubation, with relative expression values of 9.7 and 6.4, respectively. The elevated level of phosphorylated form of p38 was also detected in 5-FU-treated cells with relative expression values of 5.4 and 3.0 and in 14B-treated cells with relative expression values of 4.9 and 3.5 after 48 h and 72 h of incubation, respectively. Among the tested compounds, CX-4945 activated p38 in the lowest extent with relative expression values of 1.5 and 1.4 after 48 h and 72 h of incubation, respectively. Taking into account that p38 kinase negatively regulates cell proliferation and tumorigenesis by regulation of factors such as p53, p27, suppressor protein, p18, and Cdc25 [21], the obtained results correlated with the synergistic effect occurred in MDA-MB-231 treated with 5-FU and either 14B or CX-4945. 

Additionally, the effect of the tested drug combinations on FAK was studied. FAK is a non-receptor tyrosine kinase that is a key regulator of integrin-mediated FA (focal adhesion) assembly, and autophosphorylation of Tyr397 is the most important activator of this kinase [22]. The western blot study showed a significant decrease of FAK autophosphorylation in MDA-MB-231 treated with 5-FU plus 14B or CX-4945 with p-FAK relative expression values of 0.25 and 0.33 after 48 h and 0.04 and 0.07 after 72 h of treatment, respectively (Figure 3). This inactivation can partially be a result of the 5-FU action that decreased p-FAK significantly, with relative expression values of 0.32 and 0.12 after 48 h and 72 h of treatment, respectively. Interestingly, both 14B and CX-4945, when used separately, affected the autophosphorylation of FAK, but to a lesser extent than 5-FU alone, with relative expression values of 0.63 and 0.5 compared to control after 72 h of treatment, respectively. However, a statistically relevant difference between the phosphorylation extent of FAK in cells treated with 5-FU alone or a combination of 5-FU + 14B was detected after 72 h, which was in good agreement with the observed synergistic effect (Figure 3B). Taking into account that many studies provided strong evidence to connect FAK expression/activation to the promotion of cancer, the obtained results suggest a molecular mechanism for the synergistic interactions of 5-FU and 14B or CX-4945 in MDA-MB-231.

Since many evidences support the notion that downregulation of CK2 affects the mTOR (mammalian target of rapamycin) and ERK1/2 signaling pathways [23], we investigated the effect of 5-FU and CK2 inhibitors on ERK1/2 activity. Interestingly, all the tested compounds and combinations increased the phosphorylation of ERK1/2 after 72 h of treatment, with the strongest relative expression of p-ERK1/2 (Thr202/Tyr204) up to 4 times in cells treated with 5-FU and a combination of 5-FU with 14B. Moreover, the obtained data for 5-FU plus CX-4945 combination after both incubation times were statistically and significantly different from that of 5-FU-treated cells (Figure 3B), although an increase of phosphorylated ERK1/2 in CX-4945-treaed cells was detected only after 72 h of treatment. Enhanced phosphorylation of ERK1/2 can suggest an inhibition of the PI3K/AKT/mTORC1 and activation of ERK1/2 pathways, as it has previously been demonstrated that downregulation of protein kinase CK2 results in inhibition of the mTOR pathway, downregulation of raptor expression levels, and activation of the extracellular signaling-regulated protein kinase 1/2 (ERK1/2) signaling pathway [23].

### 2.4. Molecular Docking of 14B to the Activity Site of FAK

Molecular docking was performed to confirm the possibility of inhibition of FAK by 14B. A high-resolution (1.95 Å) crystal structure of FAK was taken from Protein Data Bank with code 4GU6 [24] and the chain A was used as a docking receptor. The non-commercial AutoDock Vina software was used to apply the molecular docking protocol [25]. 14B was docked into the active site of FAK (Figure 4A) with a binding affinity of −7.0 kcal/mol, which suggested that 14B can in fact inhibit FAK kinase. An interaction analysis in this model showed that the nitrogen atom from amino group of 14B formed a hydrogen bond (3.2 Å) with the carbonyl group of Cys502 in the hinge region (Figure 4B). Cys502 in the active site of FAK is known as a key residue for binding with ligands [26]. This interaction can be found in many X-ray crystallography structures with various inhibitors [24,27,28,29,30]. Moreover, it has been found that three bromine atoms of 14B can form halogen bonds: bromine at C4 with the carbonyl group of Ser568 (3.3 Å) and with the more distant carbonyl group of Asn551 (3.6 Å), bromine at C6 with the carbonyl group of Ile428 (3.6 Å), and bromine at C7 with the carbonyl group of Glu506 (3.7 Å). These halogen interactions can stabilize inhibitor position in the active site. Taking under consideration that molecular docking was performed with a rigid structure of the receptor, length of halogen bonds can be in fact shorter due to a small adaptation of the protein conformation in solution with the inhibitor, which can affect even better binding of 14B to FAK than that in the predicted model.

### 2.5. Confocal Laser Scanning Microscopy Examinations of MDA-MB-231 Cells

Cell migration and adhesion rely on the dynamics of the actin cytoskeleton. Therefore, we examined the organization of actin filaments of MDA-MB-231 cells after 72 h of treatment with 5-FU, 14B, and CX-4945 used separately or in combination (Figure 5). Non-treated cells were used as control. We observed that all single compounds caused changes in cell morphology, especially in the actin-rich structures, such as lamellipodia and filopodia; the changes were more visible in the cells treated with 5-FU and 14B. The cells exhibited signs of apoptotic cell death, such as chromatin condensation and blebs (Figure 5, white and red arrows, respectively). Under combinational treatment (i.e., 5-FU + CX-4945 and 5-FU + 14B), we observed even more drastic changes in MDA-MB-231; cells were characterized by the destabilization of actin network, cell rounding, and more irregular shape and exhibited features of progressing cell death.

### 2.6. Induction of Apoptosis in MDA-MB-231 Cells

In order to explain the observed synergism of 5-FU and 14B or CX-4945, induction of apoptosis in MDA-MB-231 cells was investigated by means of flow cytometry. The results showed the pro-apoptotic influence of the test compounds, used separately or in two combinations (Figure 6). The highest number of apoptotic cells was detected after treatment with the combination of 5-FU and 14B and resulted in 19% total apoptotic cells, whereas treatment with 15 µM 5-FU, 2 µM 14B, and 5 µM CX-4945 used separately resulted in 16%, 16%, and 11% total apoptotic cells. The obtained data indicated that the observed synergistic effect was poorly associated with apoptosis induction, but correlated with the anti-apoptotic role of CK2. 

Additionally, the relative expression of apoptotic markers, i.e., cleaved PARP1 (c-PARP1), caspase 7, and apoptosis inducing factor (AIF), were investigated by western blot analysis. Interestingly, we detected a statistically relevant increase of c-PARP1 in the cells treated with both the combinations compared with the cells treated with 5-FU alone (2.8 after 48 h), with the highest relative expression values of 4.8 and 7.1 for 5-FU + 14B and 5-FU + CX-4945, respectively, after 48 h of treatment (Figure 7). 

Although, after 72 h of treatment, the relative expression of c-PARP1 decreased and was statistically relevant compared to 5-FU-treated the cells with the values of 2 and 2.1 in cells treated with 5-FU + 14B and 5-FU + CX-4945, respectively. The study indicated the most significant increase of active, cleaved caspase 7 (c-caspase 7) in 5-FU-treated cells and in cells treated with the combinations. Interestingly, a significant increase in the level of both pro-form and active form of caspase 7 was detected in cells treated with 5-FU plus CX-4945 with a value of 3 after 48 h of treatment. The most significant increase of apoptogenic AIF was detected in 5-FU-treated cells with values of 3.1 and 2.3 after 48 h and 72 h of treatment, respectively, and in cells treated with the combination of 5-FU and 14B after 72 h of treatment with a relative expression value of 3.2.

Although, the results obtained for annexin V binding assay did not demonstrate significant induction of apoptosis in MDA-MB-231 after treatment with the test compounds, the relative expression of apoptotic markers in the treated cells indicated induction of apoptosis after combined treatment. Thus, these data seemed to correlate with the observed synergistic effect.

### 2.7. The Effect of the Synergistically Acting Combinations on Cell Cycle Progression in MDA-MB-231 Cells

Since it has been shown previously that the TBBi derivative [11,31], as well as 5-FU [32], can affect cell cycle progression of breast cancer cells, the influence of 5-FU in combination with either 14B or CX-4945 on cell cycle progression was tested after 48 and 72 h of treatment in MDA-MB-231 cells. The cell cycle distribution profiles after treatment with each compound used separately and in two different combinations were determined by flow cytometry. The representative plots with the calculations of cell percentages in each phase of the cell cycle are shown in Figure 8. 

The obtained results indicated that 5-FU led to S-phase arrest in MDA-MB-231 cells, with a similar number of cells after 48 h and 72 h of treatment (up to 32% more cells than in control). However, the highest number of cells in S-phase, i.e., up to 43% and 33% more than in control cells after 48 h and 72 h of treatment, respectively, was observed in cells treated with 5-FU plus CX-4945.

The observed effect correlated with the prolongation of S-phase in the cells treated with CK2 inhibitors after 48 h, when used separately, with a similar number of cells, i.e., 33% and 35% for 14B and CX-4945, respectively, resulting in 6.3% and 8.5% more cells in S-phase than in the control cells, respectively, indicating an additive interaction.

## 3. Discussion

The effect of inhibitors directed against two different molecular targets, i.e., protein kinase CK2 and TS, used alone or in combinations, was studied on two different breast cancer cell lines, MCF-7 and MDA-MB-231. Our results demonstrated that the effect of 5-FU, 14B, and CX-4945 used in combinations was dependent on the type of the cell line. Interestingly, while 5-FU plus 14B affected the viability of cells in a synergistic manner in the MDA-MB-231 cell line, it induced an antagonistic effect in the MCF-7 cell line. Otherwise, the combination of 5-FU with CX-4945 resulted in synergism in both cell lines [11]. Importantly, the obtained results indicated the possibility of decreasing an effective dose of 5-FU by about 43 or 46 times, when it is used in combination with either 14B or CX-4945, respectively. This can be extremely important as mucositis and diarrhea are common side effects of 5-FU-based anticancer regimens [33] and hormonal or anti-HER2 therapy is inefficient in TNBC. Although both the tested combinations of 14B or CX-4945 with 5-FU decreased the viability of MDA-MB-231 with a similar synergistic effect (similar CI values), 5-FU + 14B was more effective in p38 and ERK1/2 activities and the apoptosis assay than 5-FU + CX-4945. On the other hand, 5-FU + CX-4945 was more effective in c-PARP and S-phase arrest than 5-FU + 14B. The observed differences for the tested drug combinations correlated with the mode of action of single inhibitors of CK2 on the signaling pathways, i.e., 14B activated p38 MAPK stronger than CX-4945 after 48 h and 72 h of treatment and ERK1/2 after 48 h. The same correlation can be observed for the apoptosis assay and c-PARP relative expression, since 14B induced apoptosis stronger than CX-4945, although with lower relative expression of c-PARP after 48 h incubation and similar to CX-4945 after 72 h of incubation. When the tested combinations affected the cell cycle, such correlation was observed in prolongation of S-phase for both the tested combinations after 48 h of treatment. 

The obtained synergistic effect in MDA-MB-231 after combination treatment with 5-FU and either 14B or CX-4945 seemed to correlate with the inhibition of FAK and activation of p38 rather than with the direct inhibition of CK2, detected only after 48 h in cells treated with 14B alone, a combination of 14B with 5-FU, and CX-4945 alone. Since other studies demonstrated a significant decrease of phosphorylation of Ser529 in NF-κB-p65 in MDA-MB-231 cells treated with 5 µM CX-4945, the different conditions of experiments must be taken into account, as the previous data were obtained for starving cells treated with CX-4945 only for 4 h [15]. While the observed inefficient inhibition of CK2 partially may be due to an overexpression of CK2 catalytic subunits in MDA-MB-231 cells, an increase of phosphorylation extent of NF-κB-p65 in cells treated with CK2 inhibitors for 72 h was rather intriguing and difficult to understand because of the lack of correlation with either p65 total or CK2α expression. Furthermore, a significant increase of CK2-directed phosphorylation of Ser529 in NF-κB-p65 was detected in cells treated with 5-FU alone and a combination of 5-FU and CK2 inhibitors, especially after 72 h. Although there is some evidence demonstrating that 5-FU induced the upregulation of NF-κB protein expression [34], an increase of phosphorylation of Ser529 in NF-κB-p65 subunit in 5-FU-treated MDA-MB-231 cells seemed to be correlated rather with an increase of CK2 activity than with total expression of p65 as no significant differences in its relative expression were detected. Therefore, it should be taken into account that TS is a substrate for CK2 [35] and upregulated level of TS resulting from the formation of the TS-F-dUMP complex could contribute to an increase of CK2 activity. 

Interestingly, although an inefficient inhibition of CK2 was detected in MDA-MB-231 cells treated with either 14B or CX-4945, the inhibition of another kinase, i.e., FAK was indicated. Furthermore, molecular docking studies demonstrated that 14B can interact with the active site of FAK, thus confirming the results obtained by immunodetection. Our results indicating that CK2 inhibitors affect FAK are in agreement with previous data indicating inhibition of FAK by CX-4945 in A549 lung cancer cells after TGF-β1-induced activation [36]. These data demonstrated that, beyond regression of tumor mass, CX-4945 could be more suitable in a new therapy for cancer metastasis and epithelial-to-mesenchymal transition (EMT)-related disorders. The observed inhibition of FAK in MDA-MB-231 cells treated with the combinations of 5-FU and either 14B or CX-4945 indicated a synergistic effect after 72 h of incubation. Interestingly, the relative expression level of p-FAK (on Tyr397) was significantly decreased in 5-FU-treated MDA-MB-231 cells and to the best of our knowledge, this was the first study that indicated the direct effect of 5-FU on the phosphorylation extent of Tyr397 in FAK. This kinase is a widely expressed cytoplasmic protein tyrosine kinase involved in integrin-mediated signal transduction. It plays an important role in the control of several biological processes, including cell spreading, migration, and survival [37]. The activation of FAK by integrin clustering leads to its autophosphorylation at Tyr397, which is a binding site for the Src family kinases, phosphoinositide 3 kinase (PI3K) and phospholipase C-gamma (PLCγ) [38]. The recruitment of Src family kinases results in phosphorylation of Tyr407, Tyr576, and Tyr577 in the catalytic domain and Tyr871 and Tyr925 in the carboxy-terminal region of FAK. The link between FAK and breast cancers is strongly suggested by a number of reports showing that FAK gene is amplified and overexpressed in a large fraction of breast cancer specimens [39]. Recently, it was shown that, while not affecting primary tumor development and growth, FAK deletion significantly suppressed breast cancer metastasis in vivo [40]. Another study demonstrated that knockdown of FAK expression in TNBC cells or the treatment of TNBC cells with a FAK inhibitor followed by co-culture with cytokine-induced killer (CIK) cells increased the death of TNBC cells, suggesting that FAK plays an important role in sensitizing tumor cells to CIK cells [41]. Interestingly, our results supported the most recent study, which demonstrated that inhibition of FAK enhanced 5-FU chemosensitivity to gastric carcinoma [42]. Furthermore, clinical data also showed that patients with higher levels of FAK had significantly shorter overall survival and time to first progression than those with lower levels of FAK [42]. Since FAK kinase is involved in cell adhesion, and the obtained results indicated the participation of this kinase in the observed synergistic effect in MDA-MB-231 cells, we examined the organization of actin filaments both in the treated and control cells. The obtained results indicating destabilization of actin network in 5-FU-treated cells support the previous data, which demonstrated that treatment of smooth muscle cells with 5-FU induced changes in cellular and nuclear morphology [43]. Moreover, changes in the morphology of CK2 inhibitor-treated cells were observed, in spite of lack of an effective inhibition of CK2 in MDA-MB-231 cells. The results supported the evidence that CK2 affects cytoskeletal structures and correlated functions such as cell shape, mechanical integrity, cell movement, and division [44]. Furthermore, signs of apoptotic cell death, such as chromatin condensation and blebs, were observed in the treated MDA-MB-231 cells, especially under combinational treatment (i.e., 5-FU + CX-4945 and 5-FU + 14B), which were in agreement with the results demonstrating an increase expression of the apoptotic markers in those cells.Another molecular basis of the obtained synergistic effect in MDA-MB-231 treated with 5-FU and either 14B or CX-4945 was correlated with activation of p38 kinase. This mitogen-activated protein kinase (MAPK), besides its well-known functions in inflammation and other cellular stresses, including endoplasmic reticulum (ER)-induced stress, negatively regulates cell proliferation and tumorigenesis. Inactivation of the p38 pathway enhances cellular transformation and renders mice prone to tumor development with concurrent disruption of the induction of senescence. 5-FU-induced activation of p38 in MDA-MB-231 is in line with previous results demonstrating that p38 MAPK activation is a key determinant in the cellular response to 5-FU treatment [45]. The data demonstrated that inhibition of p38 MAPKα correlates with a decrease in 5-FU-associated apoptosis and chemical resistance in both HaCaT and HCT116 cells. Furthermore, the results correlate with the latest reports showing that CK2α (catalytic subunit of CK2) increases the resistance of colon cancer cells to 5-FU by inhibition of ER stress, which leads to inhibition of apoptosis and inefficacy of therapy [18]. The authors suggested that 5-FU treatment in combination with a CK2α inhibitor may exert a synergistic effect against drug-resistant cancer cells. Interestingly, other data demonstrated that 5-FU increased the expression of the ER stress marker, 78-kDa glucose-regulated protein (GRP78), in MCF-7 cells, that contributes to 5-FU resistance in breast cancer cells. The data indicated an important role of the ER stress-mediated GRP78/OCT4/lncRNA MIAT/AKT pathway in cell resistance to 5-FU, highlighting potential molecular targets for combating 5-FU resistance in breast cancer [46]. 

The observed anticancer effect in MDA-MB-231 correlated with an activation of ERK1/2 kinases, as all the tested compounds and combinations increased phosphorylation of ERK1/2 after 72 h of treatment. Our results support the notion that downregulation of protein kinase CK2 results in activation of the extracellular signaling-regulated protein kinase 1/2 (ERK1/2) signaling pathway [23].

The accumulation of cells in S-phase after 48 h and 72 h of 5-FU treatment indicated the activation of the repair mechanisms, induced by DNA damage [47]. Interestingly, the highest number of cells in S-phase, i.e., up to 43% and 33% more than in control cells after 48 h and 72 h of treatment, respectively, was observed in cells treated with 5-FU plus CX-4945. The obtained results can be partially correlated with DNA repair mechanisms as it was shown by others that CX-4945 blocked the DNA repair response induced by gemcitabine and cisplatin in ovarian cancer cells [17]. It was demonstrated that phosphorylation of MDC1, a key mediator of homologous recombination double-strand break (DSB) repair, by CK2 was necessary for the formation of a multiprotein complex that is required for DSB repair signaling [48]. Taking into account that DNA damage is an essential mechanism of cytotoxicity of 5-FU in different cell lines, inhibition of CK2 leading to blockage of DNA repair after 5-FU treatment seems to be a very likely mechanism of a synergistic interaction between 5-FU and CX-4945, similarly to the data obtained previously for MCF-7 cells [11]. In summary, the obtained results support the notion that inhibitors of TS could be more effective when combined with a drug exerting a CK2 inhibitory effect and indicate the important role of ER-induced stress in the observed anticancer synergistic effect. However, the interaction mechanism can be different and depend on the cancer cell line and drug combination; therefore, further studies are needed.

## 4. Materials and Methods 

### 4.1. Reagents and Antibodies

Dimethyl sulfoxide (DMSO), molecular biology grade, used as a solvent for all stocks of the chemical agents, was obtained from Roth (Karlsruhe, Germany). All reagents used in flow cytometry analysis were purchased from BD Biosciences Pharmingen (San Diego, CA, USA). The following primary antibodies were used: anti-p-FAK (Tyr397) (CST, #8556, 1:1000, overnight, 4 °C), anti-FAK (CST, #3285, 1:1000, overnight, 4 °C), anti-p-p38 MAPK (Thr180/Tyr182) (CST, #9211, 1:1000, overnight, 4 °C), anti-p38 MAPK (CST, #9212, 1:1000, overnight, 4 °C), anti-p-p44/42 MAPK (Erk1/2) (Thr202/Tyr204) (CST, #4094, 1:1000, overnight, +4 °C), anti-p44/42 MAPK (ERK1/2) (CST, #4695, 1:1000, overnight, 4 °C), anti-c-RARP1 (Asp214) (Cell Signaling, #9546, 1:500, overnight, 4 °C), anti-caspase 7 (CST, #8438, 1:500, overnight, 4 °C), anti-GAPDH (Merck Millipore, #MAB374, 1:20,000, 30 min, room temperature (RT)), anti-p-p65 (Ser529) (Biorbyt, #orb 14916, 1:500, overnight, 4 °C), anti-p65 (CST, #D14E12, 1:1000, overnight, 4 °C), anti-CK2α (CST, #2656, 1:1000, overnight, 4 °C), and anti-TS (Merck Millipore, #MAB4130, 1:500, overnight, 4 °C). Secondary goat anti-rabbit IgG-HRP, horseradish peroxidase (Dako, #P0448, 1:2000, 1 h, RT) and anti-mouse IgG-HRP (Dako, #P0447, 1:1000, 1 h, RT) were used in the study of the effect of drug combinations on TS, CK2α, and CK2-mediated p65 pathway. Secondary HRP-labeled antibodies, including anti-mouse (Millipore, AP308P, 1:10,000, 1 h, RT), anti-rabbit (Millipore, AP387P, 1:10,000, 1 h, RT), and anti-goat (Santa Cruz Biotechnology, sc-2020, 1:10,000, 1 h, RT) antibodies, were used in the other studies. Alexa Fluor 488 phalloidin (Invitrogen, #A12379) and Hoechst 33,342 (Life Technologies, #H3569) were used in the IF (immunofluorescence) study. Protease inhibitors (#11 836 153 001) were from Roche Applied Science (Mannheim, Germany). Nitrocellulose membrane was from GE Healthcare Life Sciences (Freiburg, Germany) and solvents for HRP reaction (WesternBright Peroxide and WesternBright Quantum) were purchased from Advansta (San Jose, CA, USA). Western ECL Substrate ECL (enhanced chemiluminescence) reagent was from Bio-Rad (Hercules, CA, USA), and CX-4945 was obtained from Biorbyt (Cambridge, UK). TBBi and its derivative 14B were obtained as described previously [19]. Other solvents, reagents, and chemicals were purchased from POCH (Avantor Performance Materials, Gliwice, Poland) Merck and Sigma-Aldrich Chemical Company (St. Louis, MO, USA). 

### 4.2. Cell Culture and Agent Treatment

Human Caucasian breast adenocarcinoma cell line, MDA-MB-231 (ECACC 92020424), was purchased from ECACC (European Collection of Authenticated Cell Cultures), whereas MCF-7 (ATCC HTB-22) was purchased from ATCC (American Type Culture Collection). MCF-7 and MDA-MB-231 were cultured in high-glucose DMEM (Lonza, Basel Switzerland) supplemented with 10% fetal bovine serum (EuroClone), 2 mM l-glutamine, and antibiotics (100 U/mL penicillin, 100 µg/mL streptomycin). MCF-7 cells were supplemented with 10 μg/mL of human recombinant insulin. Cells were grown in 75 cm^2^ cell culture flasks (Sarstedt, Nümbrecht, Germany) in a humidified atmosphere of CO_2_/air (5%/95%) at 37 °C. All the experiments were performed in exponentially growing cultures. Stock solutions of the test compounds were prepared in DMSO and stored in –80 °C for maximum one month. For the cytotoxicity studies, stock solutions of the test compounds were diluted 400-fold with the proper culture medium to obtain the final concentrations. Stock solutions were diluted in a 1:1 ratio with DMSO or the second compound for single compound and combination tests, respectively, so the final concentration of vehicle was constant in each case, and the same stock solution was used in each experiment. For cytotoxicity studies, 2-fold serial dilutions were prepared in the proper medium containing 0.5% DMSO.

### 4.3. MTT-Based Viability Assay

MCF-7 and MDA-MB-231 cells were trypsinized in 0.25% trypsin-EDTA solution (Sigma-Aldrich) and seeded onto 96-well microplates (Sarstedt) at a density of 1.5−3 × 10^4^ cells/well. After 18 h of plating, cells were treated with the test compounds or DMSO (0.5%) at the appropriate concentrations. After incubation, culture medium was discarded and MTT stock solution (Sigma-Aldrich) was added to each well to the final concentration of 1 mg/mL. After 1 h of incubation at 37 °C, water-insoluble dark blue formazan crystals were dissolved in DMSO (200 µL) during incubation for 10 min at 37 °C. Optical densities were measured at 570 nm using the BioTek (Winooski, VT, USA) microplate reader. All measurements were carried out in a minimum of six replicates and the results were expressed as the fraction of not viable cells (Fa) relative to control (cells without inhibitor in 0.5% DMSO). Fa values were calculated from the equation, 1–(T–B/C–B), where T and C are absorbances obtained for the treated and untreated cells, respectively, whereas B refers to the absorbance of blank well (without cells). The data were analyzed on the basis of the Chou-Talalay model for synergistic interactions, using the CalCusyn software (BIOSOFT, Cambridge, United Kingdom) [20,49].

### 4.4. Cell Cycle Analysis

MDA-MB-231 cells were cultured in 25 cm^2^ culture flasks and treated with the test compounds as described above. After exposure to the test compounds, the cells were trypsinized, collected, and washed with cold PBS (phosphate buffered saline) and fixed at −20 °C in 70% ethanol for at least 24 h. Subsequently, the cells were washed in PBS and stained with 50 μg/mL PI (propidium iodide) and 100 μg/mL RNase solution in PBS supplemented with 0.1% *v/v* Triton X-100 (Roth, Karlsruhe, Germany) for 30 min in the dark at RT. Cellular DNA content was determined by flow cytometry employing the FACSCanto II flow cytometer (BD Biosciences, San Jose, CA, USA), and analyzed using the BD FACSDiva and WinMDI 2.8 software package written by Joe Trotter of the Scripps Institute (La Jolla, CA, USA). The DNA histograms obtained were analyzed using the MacCycle software (Phoenix Flow Systems, San Diego, CA, USA) for evaluation of distribution of cells in different phases of the cell cycle.

### 4.5. Detection of Apoptosis

MDA-MB-231 cells were cultured in 6-well plates and treated with the test compounds as described above. Subsequently, the cells were trypsinized, collected by centrifugation at 200× *g* at 4 °C for 5 min, washed twice in cold PBS, and subsequently suspended in binding buffer at 1 × 10^6^ cells/mL. Then, 100 μL aliquots of the cell suspension were labeled according to the kit manufacturer’s instructions. Briefly, annexin V-FITC and PI (BD Biosciences Pharmingen, San Diego, CA, USA) were added to the cell suspension, and the mixture was vortexed and incubated for 15 min at RT in the dark. Then, 400 μL of cold binding buffer was added, and the cells were vortexed again and kept on ice. Flow cytometry measurements were performed within 1 h after the labeling. Viable, necrotic, early, and late apoptotic cells were detected by flow cytometry using BD FACSCanto II (BD Biosciences, San Jose, CA, USA) and analyzed with the BD FACSDiva software.

### 4.6. Western Blotting

The MDA-MB-231 cells growing exponentially were seeded at 4.8 × 10^5^ cells in 6 cm diameter plates. Subsequently, compounds were added in a final concentration of 0.5% DMSO. After incubation, cell monolayer was washed three times in ice-cold PBS and the cells were scraped and lysed in RIPA buffer (50 mM Tris-HCl (pH 7.4), 1% NP-40, 0.5% sodium deoxycholate, 0.1% SDS, 150 mM NaCl, 2 mM EDTA, 50 mM NaF, 0.2 mM sodium orthovanadate, and protease inhibitors cocktail; Roche). Subsequently, cells were sonicated (3 times for 15 s), centrifuged at 20,000× *g* for 15 min at 4 °C, and supernatants were collected and stored at −80 °C. The protein concentration was determined using Bradford assay. Then, supernatants were incubated with the Laemmli buffer for 5 min at 98 °C. Equivalent amounts of total protein (10–20 μg) were analyzed by SDS-PAGE and subsequently western blotting was performed using primary antibodies in blocking buffer containing either 3% BSA (bovine serum albumin) in TBST, Tris-buffered saline, 0.1% Tween 20 (10 mM Tris-HCl (pH 8), 150 mM NaCl, 0.1% Tween 20) to study the effect of drug combinations on TS, CK2α, and CK2-mediated p-65 pathway or 3–5% milk or BSA in TBS (Tris-buffered saline) with 0.2% Triton X-100 in other western blot studies. After overnight incubation at 4 °C with the primary antibodies, the membranes were washed with TBST and probed with the respect secondary antibodies. The ECL substrate (Bio-Rad or Millipore) was used for detection and immunoblots were scanned using G Box Chemi (Syngene, Cambridge, UK) with GeneSys software (Syngene, Cambridge, UK).

### 4.7. Densitometry

For densitometry, immunoblots were scanned using G Box Chemi (Syngene) or medical X-ray blue/MXBE films developed at Fuji and the FPM 800A developing machine, and the density of each lane of phosphorylated and total protein was quantified using Image J software. Phosphorylated protein densities were normalized to GAPDH densities, assuming 100% for untreated cells and then they were converted to a percent of the appropriate control.

### 4.8. Statistical Evaluation

Results were represented as mean ± SEM of at least three independent experiments performed in triplicate. The statistical analysis was performed using the GraphPad Prism 5.0 software (GraphPad Software Inc., San Diego, CA, USA). Significance was determined using a *t*-test. The statistical significance of differences was indicated in figures by asterisks as follows: * *p* ≤ 0.05, ** *p* ≤ 0.01, and *** *p* ≤ 0.001.

### 4.9. Immunocytochemical Staining and Microscopy Analysis

Cells were seeded in 4-well dishes (35/10 mm; Greiner Bio-One North America, Inc., Monroe, NC, USA) and cultured in appropriate conditions. After 18 h of culturing, cells were treated with 15 µM 5-FU, 5 µM CX-4945, or 2 µM 14B alone or in the following combinations: CX-4945 + 5-FU (5 + 15 µM), 14B + 5-FU (2 + 15 µM) for up to 72 h. Subsequently, cells were washed twice with PBS, fixed with 3.7% paraformaldehyde (PFA) solution for 20 min, washed twice with PBS, then quenched for 30 min with 50 mM NH4Cl, and finally incubated for 1.5 h in a blocking solution (2% horse serum in PBS/0.02% Triton X-100). To visualize actin filaments, cells were stained for 20 min with Alexa Fluor 488-conjugated phalloidin (diluted 1:200 in PBS) and then washed three times with PBS/0.02% Triton X-100. Then, cells were extensively washed in PBS/0.02% Triton X-100 and stained with the Olympus FLUOROVIEW FV1000 confocal laser scanning microscope (CLSM, Olympus, Center Valley, PA, USA). 

### 4.10. Molecular Docking

Molecular docking was carried out using 1.1.2. AutoDock Vina program (The Scripps Research Institute, Molecular Biology, La Jolla, CA, USA) [25]. All ligands were drawn in MarvinSketch (ChemAxon, Budapest, Hungary) and saved as mol2 files. The hydrogen atoms and Gasteiger partial charges were added by 1.5.6. AutoDock tools (The Scripps Research Institute, Molecular Biology, La Jolla, CA, USA) [30] and the ligand files were saved in pdbqt format. The crystal structure of FAK kinase was taken from Protein Data Bank with the PDB code, 4GU6 chain A [24]. All water molecules and inhibitors were removed, the polar hydrogens were added, and Gasteiger charges were calculated using AutoDock tools to get files in pdbqt format. AutoGrid was used to find the appropriate grid box size. The box center was set at 25.000, −10.000, and −40.000 (*x*, *y*, and *z* coordinates, respectively) and the final size space dimensions were x = 40 Å, y = 40 Å, and z = 40 Å. Docking was performed with an exhaustiveness level of 32.

## Figures and Tables

**Figure 1 ijms-21-06234-f001:**
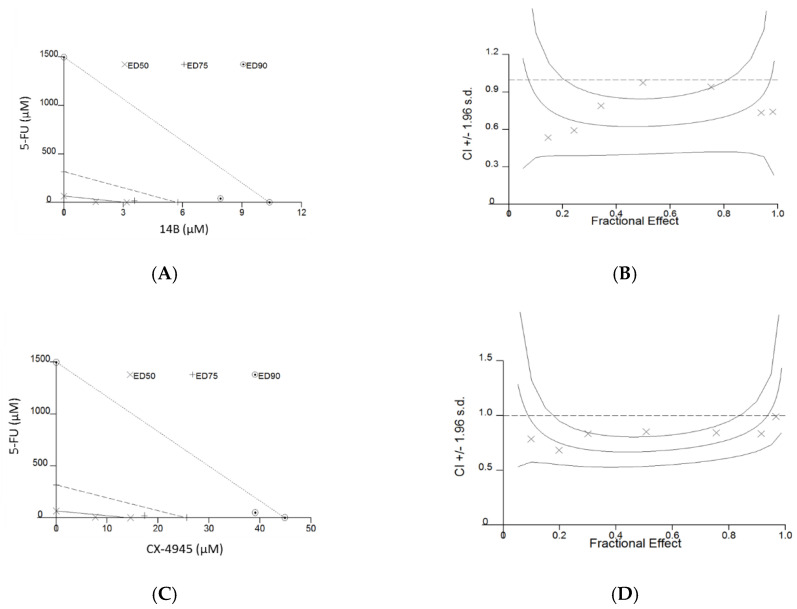
Synergistic interaction of 5-fluorouracil (5-FU) with CK2 inhibitors, i.e., 14B or CX-4945, in inhibition of viability of the MDA-MB-231 cell line. (**A**,**C**) Representative CalcuSyn software-simulated isobolograms. (**B**,**D**) Fraction affected-combination index (Fa-CI) curves for the respect combinations of 5-FU with either 14B or CX-4945 after 72 h of treatment, generated after fitting Fa obtained from the MTT assay.

**Figure 2 ijms-21-06234-f002:**
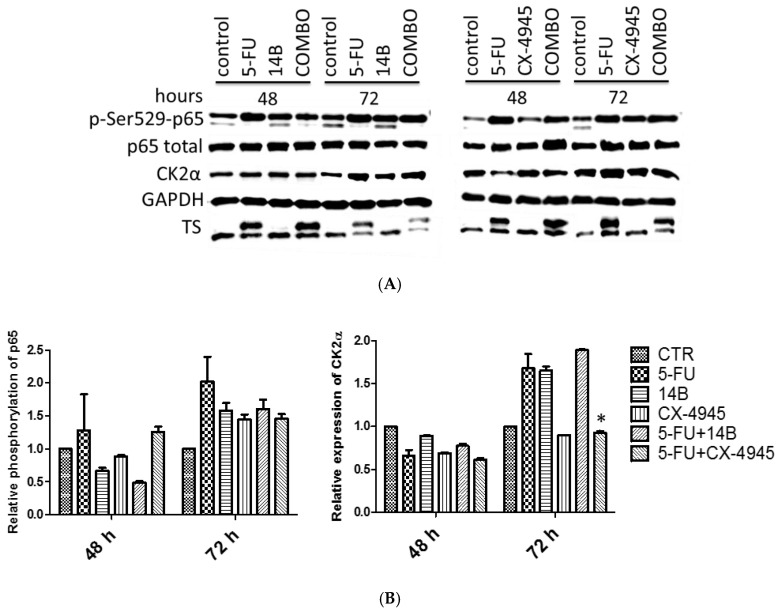
Western blot analysis of thymidylate synthase (TS), protein kinase CK2α subunit, and total nuclear factor kappa-light-chain-enhancer of activated B-cells p-65 subunit (NF-ĸB-p65 and p-Ser529-p65) in the crude extracts obtained from MDA-MB-231 cells after 48 h and 72 h of treatment with 15 µM 5-FU and 2 µM 14B, used separately and in combination (left panels), as well as 15 µM 5-FU and 5 µM CX-4945 used separately and in combination (right panels). GAPDH (glyceraldehyde 3-phosphate dehydrogenase) was used as a loading control for each sample. Preparation of cell extracts and protein detection are described in the Materials and Methods: (**A**) The representative blots; (**B**) densitometry quantifications for p-p65 and CK2α are shown in the graphs, with untreated cells serving as the reference point (1). * Statistically, significantly different from that of 5-FU-treated cells and the drug combinations by student’s *t*-test (*p* ≤ 0.05) (see Materials and Methods).

**Figure 3 ijms-21-06234-f003:**
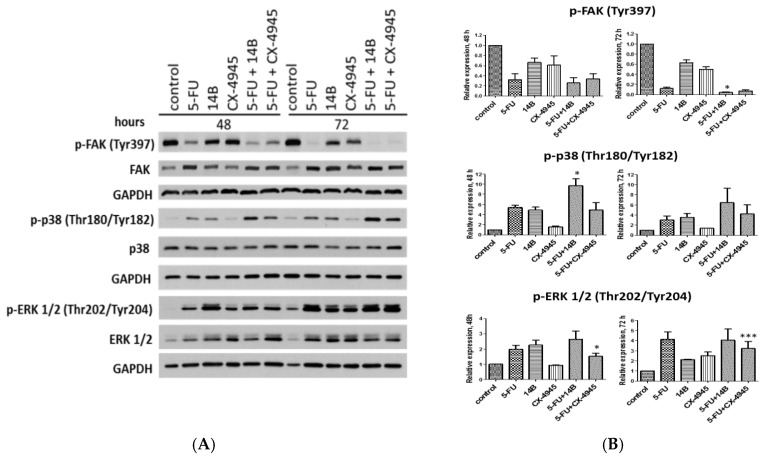
Western blot analysis of FAK, p38, and ERK1/2 kinases in crude extracts obtained from MDA-MB-231 after 48 h and 72 h treatments with 15 µM 5-FU, 2 µM 14B, and 5 µM CX-4945 used separately or in combination. GAPDH was used as a loading control for each sample: (**A**) The representative blots; (**B**) densitometry quantifications for phosphorylated proteins are shown in the graphs, with untreated cells serving as the reference point (1). Statistically, significantly different from that of 5-FU-treated cells and the drug combinations by student’s *t*-test were indicated in figures by asterisks as follows: * *p* ≤ 0.05, ** *p* ≤ 0.01, and *** *p* ≤ 0.001. (see Materials and Methods).

**Figure 4 ijms-21-06234-f004:**
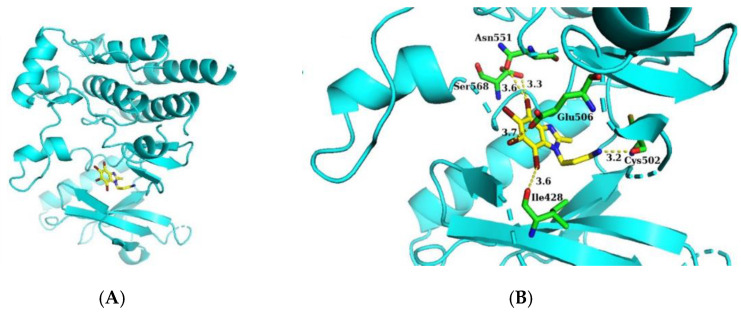
Binding model of 14B (in yellow) in the active site of FAK kinase: (**A**) Overall view; (**B**) details of binding.

**Figure 5 ijms-21-06234-f005:**
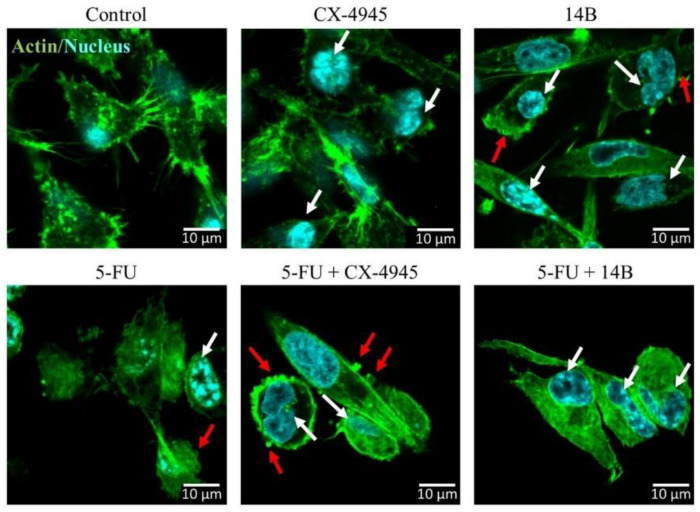
Confocal laser scanning microscopy examinations of MDA-MB-231 human breast cancer cells, untreated (control) or treated with 15 µM 5-FU, 2 µM 14B, and 5 µM CX-4945 used separately or in combinations for up to 72 h, then fixed and stained with Alexa Fluor 488 phalloidin (F-actin, green fluorescence) and Hoechst 342 (nuclei, cyan fluorescence). White arrows, abnormal cell nuclei (irregular nuclei with condensed chromatin); red arrows, blebs.

**Figure 6 ijms-21-06234-f006:**
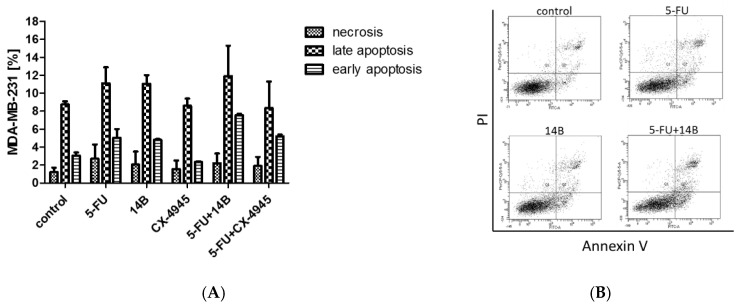
Pro-apoptotic activity of 5-FU, 14B, and CX-4945 on MDA-MB-231, used separately and in combinations. The data were determined by FACSCanto II flow cytometer (fluorescence-activated cell sorter) after 72 h treatment. Cells were stained with annexin V-FITC and PI (propidium iodide). (**A**) Mean and standard deviation (SD) of necrosis, and early and late apoptosis as percent from three independent experiments each; (**B**) representative cytograms for control (0.5% DMSO, dimethyl sulfoxide), 15 µM 5-FU, 2 µM 14B, and a combination of 5-FU and 14B.

**Figure 7 ijms-21-06234-f007:**
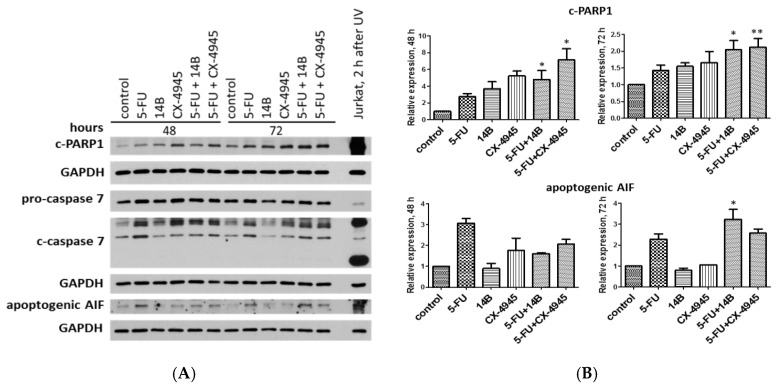
Western blot analysis of cleaved PARP1 (c-PARP1), the pro-form and active form of caspase 7 (pro-caspase 7 and c-caspase 7, respectively), and the apoptogenic form of apoptosis inducing factor (AIF) in crude extracts obtained from MDA-MB-231 cells after 48 h and 72 h of treatment with 15 µM 5-FU, 2 µM 14B, and 5 µM CX-4945, used separately or in combination. GAPDH was used as a loading control for each sample. (**A**) The representative blots; (**B**) densitometry quantifications for c-PARP1 and apoptogenic AIF are shown in the graphs, with untreated cells serving as the reference point (1). * Statistically, significantly different from that of 5-FU-treated cells and the drug combinations by student’s *t*-test (*p* ≤ 0.05) (see Materials and Methods). Extract from UV-treated Jurkat cells was used as a positive control. After 2 h, one-minute UV-irradiated Jurkat leukemic T-cell line was lysed and subjected to analysis.

**Figure 8 ijms-21-06234-f008:**
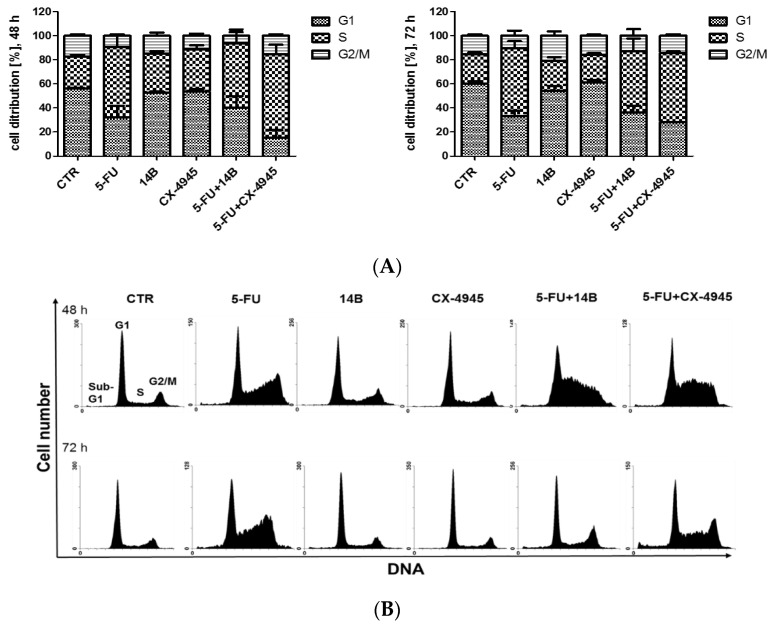
The effect of compound combinations on cell cycle progression in the MDA-MB-231 cell line. Cells were treated with 5-FU, 14B, or CX-4945, used separately or in combination for 48 h or 72 h, and then fixed and stained with PI. (**A**) Distribution of the cells in different phases of the cell cycle was determined by flow cytometry and analyzed with the MacCycle software to determine the percentage of cells in each phase of the cell cycle; (**B**) exemplary DNA histograms of the MDA-MB-231 cell line shown using the WinMDI 2.8 software.

**Table 1 ijms-21-06234-t001:** The drug potency (*D_m_*) obtained after fitting the MTT-based assay data to the median effect equation using the CalcuSyn Software.

Compound	Molecular Target	*D_m_* * ± SD (µM)	
		MDA-MB-231	MCF-7 **
5-Fluorouracil (5-FU)	TS	61.84 ± 7.64	14.88 ± 2.56
4,5,6,7-tetrabromo-1-(3-bromopropyl)-2-methyl-1*H*-benzimidazole (14B)	CK2	3.94 ± 1.08	4.28 ± 0.56
CX-4945 (silmitasertib)	CK2	12.47 ± 2.99	8.36 ± 0.35

* *D_m_* values were obtained after fitting the MTT-based assay data to median effect equation using the CalcuSyn software; ** the data for 5-FU and CX-4945 were obtained previously [11].

**Table 2 ijms-21-06234-t002:** Combination index (CI) calculated at effective doses ED_50_, ED_75_, and ED_90_, drug potency (*D_m_*), and dose reduction index (DRI) for the tested drug combinations for MDA-MB-231 and MCF-7 cells after 72 h treatment. CI values were generated by the CalcuSyn software after fitting fraction affected (Fa) values obtained by the MTT-based assay.

Cell Line	Drugs in Combination	Combination Index at			*D_m_*	DRI for 5-FU at Fa = 0.95	Type of Drug Interaction **
		ED_50_	ED_75_	ED_90_			
**MDA-MB-231**	5-FU:14B	0.69 ± 0.11	0.66 ± 0.09	0.76 ± 0.14	3.89 ± 0.54	43.42	Synergism
	5-FU:CX-4945	0.70 ± 0.06	0.65 ± 0.10	0.64 ± 0.23	11.93 ± 2.54	46.27	Synergism
**MCF-7**	5-FU:14B	2.94 ± 0.48	1.16 ± 0.39	2.38 ± 0.54	1.86 ± 1.15	-	Antagonism
	5-FU:CX-4945 *	0.92 ± 0.11	0.74 ± 0.07	0.62 ± 0.08	11.31 ± 3.68	3.25	Synergism

* Data were obtained previously [11]; ** type of interaction according to Chou [20].

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
