# Peer review of "Synergistic Interactions of 5-Fluorouracil with Inhibitors of Protein Kinase CK2 Correlate with p38 MAPK Activation and FAK Inhibition in the Triple-Negative Breast Cancer Cell Line"

_ijms, 2020, doi:10.3390/ijms21176234_

Round 1

Reviewer 1 Report

Wińska et al. performed combination treatments of 5-FU and inhibitors of protein kinase CK2. The authors show the effects of these drugs on cell proliferation, survival, morphology and intracellular signaling activities in breast cancer cells. The authors found that 5-FU can inhibit FAK phosphorylation. This manuscript may contribute to breast cancer therapy. However, some points should be addressed before publication.

<Comments>

  1. The authors should show the results of statistical analyses in Figure 2b.

  1. In page 7 line 247, the authors mention "Enhanced phosphorylation of ERK1/2 can suggest an inhibition of the PI3K/AKT/mTORC1 and activation of ERK1/2 pathways.". The authors should explain the reason why "an inhibition of the PI3K/AKT/mTORC1" can be suggested. Do the authors have experimental data about this?

  1. In Figure 5, quantification and statistical analyses of the frequencies of abnormal nuclei and bleb appearance are needed.

  1. What is the criteria to determine abnormal nuclei (chromatin condensation)?

  1. The authors show the results of treatments, 5-FU+14B and 5-FU+CX-4945. 5-FU+14B is more effective in p38 and ERK1/2 activities (Figure 3), and apoptosis assay (Figure 6) than 5-FU+CX-4945. On the other hand, 5-FU+CX-4945 is more effective in c-PARP (Figure 7) and S phase arrest (Figure 8) than 5-FU+14B. The authors should discuss this difference in the manuscript.

  1. In page 16, the authors mention about anti-vinculin. Where the authors show the results of vinculin staining?

Author Response

  1. The authors should show the results of statistical analyses in Figure 2b.

Answer: A statistical analysis was carried out, the figure 2b was corrected and the following sentence was added to the description of the figure: „*Statistically significantly different from that of 5-FU treated cells and the drug combinations by t-student test (p≤0.05) (see Materials and Methods).

2. In page 7 line 247, the authors mention "Enhanced phosphorylation of ERK1/2 can suggest an inhibition of the PI3K/AKT/mTORC1 and activation of ERK1/2 pathways.". The authors should explain the reason why "an inhibition of the PI3K/AKT/mTORC1" can be suggested. Do the authors have experimental data about this?

Answer: Unfortunately, it was not studied by us, but it was demonstrated by others that enhanced phosphorylation of ERK1/2 suggests an inhibition of the PI3K/AKT/mTORC1 and activation of ERK1/2 pathways (reference 23). In order to clarify an additional sentence has been added:

“(…), as it has previously been demonstrated that downregulation of protein kinase CK2 results in inhibition of (mTOR) pathway, downregulation of Raptor expression levels and activation of the extracellular signaling-regulated protein kinase 1/2 (ERK1/2) signaling pathway.”

3. In Figure 5, quantification and statistical analyses of the frequencies of abnormal nuclei and bleb appearance are needed.

Answer: In Figure 5 we showed the results of morphological analysis of examined cells, that was an qualitative experiment of the cell death study. This morphological technique helped us to observe the morphological changes that occur during apoptosis, such as cell shrinkage and blebbing of the plasma membrane, condensation of nuclear chromatin, and the karyorrhexis. For the quantification of a number of apoptotic cells we used other more specific technique FACS (see Figure 6).

4. What is the criteria to determine abnormal nuclei (chromatin condensation)?

Answer: Under control normal conditions the nuclei of MDA-MB-231 cells are mainly oval. An irregular and cell nuclei with condensed chromatin were indicated as abnormal. The supplementing information was added to the description of the Figure 5 (page 9): …(irregular nuclei with condensed chromatin)

5. The authors show the results of treatments, 5-FU+14B and 5-FU+CX-4945. 5-FU+14B is more effective in p38 and ERK1/2 activities (Figure 3), and apoptosis assay (Figure 6) than 5-FU+CX-4945. On the other hand, 5-FU+CX-4945 is more effective in c-PARP (Figure 7) and S phase arrest (Figure 8) than 5-FU+14B. The authors should discuss this difference in the manuscript.

Answer: The following sentence was added in the discussion (page 12):

„Although both tested combinations of 14B or CX-4945 with 5-FU decreased viability of MDA-MB-231 with the similar synergistic effect (similar CI values), 5-FU+14B is more effective in p38 and ERK1/2 activities (Figure 3), and apoptosis assay (Figure 6) than 5-FU+CX-4945. On the other hand, 5-FU+CX-4945 is more effective in c-PARP (Figure 7) and S phase arrest (Figure 8) than 5-FU+14B. The observed differences for the tested drug combinations correlate with the mode of action of single inhibitors of CK2 on the signaling pathways, i.e. 14B activates p38MAPK stronger than CX-4945 after 48 h and 72 h of treatment and ERK1/2 after 48 h. The same correlation can be observed for apoptosis assay and c-PARP relevant expression, since 14B induced apoptosis stronger than CX-4945, although with the lower relevant expression of c-PARP after 48 h incubation and similar to CX-4945 after 72 h of incubation. When the tested combinations affect the cell cycle, such correlation can be observed in prolongation of S-phase for both tested combinations after 48 h of treatment”

6. In page 16, the authors mention about anti-vinculin. Where the authors show the results of vinculin staining?

Answer: It was a mistake, and the following sentence has been removed: „Next, cells were incubated with primary antibodies (anti-vinculin, 1:1000) overnight at 4°C and subsequently cells were washed and incubated with Alexa Fluor 546-conjugated anti-mouse diluted 1:500 (ThermoFisher Scientific, USA) for 1.5h at RT.”

Reviewer 2 Report

In this manuscript, Wińska et al. describe the synergistic interactions of 5-Fluorouracil with Inhibitors of Protein Kinase CK2 in a Triple Negative Breast Cancer Cell Line, MDA-MB-231. This is a well conducted study and the results are clearly presented.  Cellular, molecular biologic and binding modeling studies were conducted and the data analyzed on the basis of the Chou-Talalay model for synergistic interactions. The authors may provide additional examples of this analytical model in breast cancer drug studies, such as the following citation: Nair et al., Cancer Letters 250 (2), 311-322, 2006.

Author Response

Protein Kinase CK2 in a Triple Negative Breast Cancer Cell Line, MDA-MB-231. This is a well conducted study and the results are clearly presented.  Cellular, molecular biologic and binding modeling studies were conducted and the data analyzed on the basis of the Chou-Talalay model for synergistic interactions. The authors may provide additional examples of this analytical model in breast cancer drug studies, such as the following citation: Nair et al., Cancer Letters 250 (2), 311-322, 2006. 

Answer: The following sentence and the suggested reference [49] were added in Materials and Methods (page 15, section 4.3):

“The data were analyzed on the basis of the Chou-Talalay model for synergistic interactions, using CalCusyn software [20, 49].”

Round 2

Reviewer 1 Report

The authors have answered now all questions sufficiently.